# Voltage-Gated Sodium Channel Na_V_1.5 Controls NHE−1−Dependent Invasive Properties in Colon Cancer Cells

**DOI:** 10.3390/cancers15010046

**Published:** 2022-12-22

**Authors:** Osbaldo Lopez-Charcas, Lucile Poisson, Oumnia Benouna, Roxane Lemoine, Stéphanie Chadet, Adrien Pétereau, Widad Lahlou, Serge Guyétant, Mehdi Ouaissi, Piyasuda Pukkanasut, Shilpa Dutta, Sadanandan E. Velu, Pierre Besson, Driffa Moussata, Sébastien Roger

**Affiliations:** 1EA4245, Transplantation, Immunologie et Inflammation, Faculté de Médecine, Université de Tours, 37032 Tours, France; 2Service D’anatomie et de Cytologie Pathologiques, Hôpital Trousseau, CHU de Tours, 37170 Tours, France; 3Service D’hépato-Gastroentérologie et de Cancérologie Digestive, Hôpital Trousseau, CHU de Tours, 37170 Tours, France; 4Service de Chirurgie Viscérale et Oncologique, Hôpital Trousseau, CHU de Tours, 37170 Tours, France; 5Department of Chemistry, University of Alabama at Birmingham, Birmingham, AL 35294-1240, USA

**Keywords:** colon cancer, Na_V_ channels, cell invasiveness, small-molecule inhibitors, proton efflux

## Abstract

**Simple Summary:**

Colorectal cancer is a major cause of morbidity and mortality that affects men and women equally. Here, we present the role of voltage-gated sodium channels (Na_V_) and Sodium/hydrogen exchanger type 1 (NHE-1) in conferring invasive capabilities to colorectal cancer cells. We also study how pharmacological inhibition or suppression of their expression by siRNA modifies the aggressive cancer cell phenotype. Our findings suggest that Na_V_ channels and NHE-1 are pertinent molecular targets for cancer therapy.

**Abstract:**

Colorectal cancer (CRC) is the second leading cause of death worldwide, with 0.9 million deaths per year. The metastatic stage of the disease is identified in about 20% of cases at the first diagnosis and is associated with low patient-survival rates. Voltage-gated sodium channels (Na_V_) are abnormally overexpressed in several carcinomas including CRC and are strongly associated with the metastatic behavior of cancer cells. Acidification of the extracellular space by Na^+^/H^+^ exchangers (NHE) contributes to extracellular matrix degradation and cell invasiveness. In this study, we assessed the expression levels of pore-forming α-subunits of Na_V_ channels and NHE exchangers in tumor and adjacent non-malignant tissues from colorectal cancer patients, CRC cell lines and primary tumor cells. In all cases, *SCN5A* (gene encoding for Na_V_1.5) was overexpressed and positively correlated with cancer stage and poor survival prognosis for patients. In addition, we identified an anatomical differential expression of *SCN5A* and *SLC9A1* (gene encoding for NHE-1) being particularly relevant for tumors that originated on the sigmoid colon epithelium. The functional activity of Na_V_1.5 channels was characterized in CRC cell lines and the primary cells of colon tumors obtained using tumor explant methodologies. Furthermore, we assessed the performance of two new small-molecule Na_V_1.5 inhibitors on the reduction of sodium currents, as well as showed that silencing *SCN5A* and *SLC9A1* substantially reduced the 2D invasive capabilities of cancer cells. Thus, our findings show that both Na_V_1.5 and NHE-1 represent two promising targetable membrane proteins against the metastatic progression of CRC.

## 1. Introduction

Colorectal cancer (CRC) is classified as the second most deadly malignant cancer worldwide representing about 10% of all cancer-related deaths. Projections based on aging, population growth, human development and particularly a diet that is increasingly westernized predict a global burden of new CRC cases that will reach about 3 million by 2040 [1]. Metastatic disease is identified in one out of five CRC patients at diagnosis and organs such as the liver, lungs and peritoneum are commonly affected [2]. After the dissemination of cancer cells to distant organs and the apparition of metastases, CRC is in a late and often incurable stage. The acquisition of an aggressive cancer phenotype not only relies on the accumulation of diverse mutated oncogenes or tumor-suppressor genes but also on the expression of genes that are silenced in differentiated tissues but aberrantly deregulated during cancer transformation [3]. Early detection and resection of pre-neoplastic lesions and low-grade tumors of the intestinal epithelium are critical to improving patient survival outcomes. However, we still lack reliable diagnostic biomarkers, which could be accessed from the intestinal lumen using endoscopic examinations. Furthermore, the absence of molecular targets whose pharmacological intervention can reduce or prevent metastasis is also a pending debt.

Voltage-gated sodium (Na_V_) channel proteins, composed of one main pore-forming α-subunit which interacts with one or two auxiliary β-subunits, are responsible for the triggering and propagation of action potentials in excitable tissues [4]. However, the proteins of Na_V_ α- and β-subunits are aberrantly expressed in virtually all human carcinomas including the most deadly malignancies such as those originating in colon, breast and lung epithelia [5,6,7]. The expression and activity of Na_V_ sodium channels in cancer cells have been associated with cancer progression, mainly by promoting the migratory and invasive abilities of cancer cells in both in vitro and in vivo models [8,9].

In the last three decades, studies have shed light on the signaling pathways and mechanisms involving these membrane proteins in cancer cells which, for β-subunits, concern enhanced vascularization and tumor growth, the promotion of a bipolar cell morphology and a hybrid aggressive mesenchymal–amoeboid phenotype, but also a reduction of cell adhesion to the substrate, cell-to-cell contacts and apoptosis [7,10,11]. On the other hand, the mechanisms described for the α-subunits in cancer include the dysregulation of sodium homeostasis, depolarization of the cell membrane, F-actin polymerization, invadopodial formation, Na^+^/H^+^ exchanger type 1 (NHE-1) allosteric modulation and enhanced secretion and activity of extracellular matrix proteases [7,12,13,14,15,16,17]. All of these pro-tumor mechanisms are crucial in providing cancer cells with an aggressive and difficult-to-treat phenotype. However, the fact that their promotion depends, at least in part, on the expression and activity of Na_V_ sodium channels represents a promising opportunity for these proteins to be used as pertinent therapeutic targets and markers of malignancy, with the advantage that they can be easily accessed from the intestinal lumen.

Particularly for colon cancer, it has been found that the *SCN5A* gene encoding for the Na_V_1.5 channel is functionally expressed in colon cancer cell lines and that its pharmacological and transcriptional inhibition decrease cell invasiveness [18,19]. In addition, the Na_V_1.5 channel protein is abundantly expressed in tumor tissue whereas it is scarce or absent in normal samples. Furthermore, the *SCN5A* gene acts as a key regulator of signaling pathways that promote invasiveness in colon cancer [20]. However, to date, it is not known whether there is differential expression of Na_V_1.5 channels and NHE-1 depending on tumor location, nor is there evidence of Na_V_ channel activity in primary tumor cells. Furthermore, the role of Na_V_ channels in modulating NHE-1 to promote invasive capabilities as well as the relevance of their inhibition with new small-molecule drugs as an anti-invasive treatment have not been studied to date.

In this study, we explored the differential expression of *SCN5A* (Na_V_1.5) and *SLC9A1* (NHE-1) genes depending on the anatomical location of tumors on colon epithelium. We show that the overexpression of *SCN5A* is positively correlated with cancer stage and poor survival for cancer patients. We also identified that the Na_V_1.5 channel is the main isoform expressed and responsible for generating macroscopic Na^+^ currents in the CRC cell lines HCT116, HT29, SW480 and SW620, as well as in colon tumor-derived primary cells (CTDPC) obtained using explants from the tissue of CRC patients. We show that NHE-1 is an important regulator of the H^+^ efflux in these cells and a major actor in the mechanisms of the invasiveness of CRC. Thus, our findings show that both Na_V_1.5 and NHE-1 contribute to cancer cell invasiveness, therefore representing promising molecular targets against metastatic disease.

## 2. Materials and Methods

### 2.1. Ethics Statement

This research was approved by the Scientific and Ethics Committees of the University Hospital of Tours and was performed in accordance with the ethical principles described in the 1964 Declaration of Helsinki. CRC tumors were kept at the hospital’s Tumor Biobank (approval number: DC2008-308).

### 2.2. Human Biopsies

Informed written consent was obtained from all the participants prior to their inclusion in this study. A total of 136 biopsies of colorectal cancer paired with normal adjacent tissue were included in this study. Human biopsies were classified by anatomical location of tumor as follows: 24 from caecum, 43 from right colon, 14 from left colon and 55 from sigmoid colon (Appendix A). All samples were obtained via colectomies at the Department of Hepato-Gastroenterology and Digestive Oncology as well as at the Department of Visceral Surgery and Oncology of the Trousseau Hospital of Tours. All patients were histopathologically diagnosed and cancer stage was determined according to the TNM staging system. After sampling, patients were referred for specialized treatments based on the guidelines of the Institute National du Cancer (INCa, Paris, France).

### 2.3. RNA Extraction, Reverse Transcription and Real-Time PCR

Colon biopsies from human bowel tissue were sampled during the endoscopic intervention procedure. Resected samples were immediately frozen in liquid nitrogen and stored at the Biological Resource Center of CHU of Tours. Then, 10–20 µm slices of tissue from selected samples were deposited in 2 mL Precellys^®^ tubes (Bertin-Instruments, Montigny-le-Bretonneux, France) containing 1.4 mm beads and 1 mL of TRIzol™ (Thermo Fisher Scientific; Waltham, MA, USA). Human tissues were disrupted using a Precellys^®^ Evolution Super Homogenizer (Bertin-Instruments, Montigny-le-Bretonneux, France) by 3 cycles of 25 s each at 6800 rpm with liquid nitrogen cooling. Total RNA isolation was performed according to the TRIzol™ recommendation protocol. High-purity total RNA from cell lines was obtained using the NucleoSpin^®^ RNA plus kit (Macherey-Nagel, Dueren, Germany) following the manufacturer’s instructions. RNA yield and purity were determined via spectrophotometry using a NanoDrop^TM^ One (Thermo Fisher Scientific; Waltham, MA, USA) and only those samples with A_260_/A_280_ and A_260_/A_230_ ratios above 1.9 were kept for further experiments. Reverse transcription for 1 µg of total RNA was carried out using the PrimeScript™ RT reagent Kit (Takara, Shiga, Japan) in 20 µL of final volume according to the manufacturer’s specifications. Gene expression of voltage-gated sodium channels (*SCN1A-SCN11A*) and major plasma membrane Na^+^/H^+^ exchangers type 1 to 4 (*SLC9A1-SLC9A4*) were assessed in all samples via quantitative RT-PCR using validated primers (Appendix A) and the kit TB Green^®^ Premix Ex Taq™ (Takara, Shiga, Japan). Genes for *HPRT1* and *PPIA* were used as reference genes. Experiments were run in triplicates in a final volume of 10 µL, including 50 ng of cDNA template, 5 µL of TB Green^®^ Premix Ex Taq™ master mix, 0.5 µL of each primer (4 µM) and 3 µL of water. Cycling was carried out using a LightCycler^®^ 480 Instrument II (Roche, Meylan, France) with the following conditions: a PCR activation step at 50 °C for 2 min followed by 95 °C for 2 min, then 40 cycles of melting at 95 °C for 15 s, annealing at 60 °C for 30 s and extension at 72 °C for 10 s. Gene expression levels were normalized in each sample to the expression of the internal reference gene and relativized to the normal condition; results are presented as fold-change values calculated using the comparative 2^−ΔΔ*CT*^ method.

### 2.4. Colorectal Cancer Gene Array and RNA Sequencing Datasets

In this work, we made use of datasets from the NCBI Gene expression Omnibus Repository, The Cancer Genome Atlas (TCGA) and EMBL-EBI Expression Atlas to explore the expression of *SCN5A* and *SLC9A1* genes in normal, tumor and metastatic tissue, expression according to cancer stage, overall survival analysis and gene expression based on tumor anatomical location. The expression of *SCN5A* and *SLC9A1* genes was assessed using the Affymetrix HGU133 Gene Array platform (NCBI Gene expression Omnibus Repository) of a cohort composed of 377 normal tissues, 1450 tumors and 99 metastatic colon biopsies; data was processed using the TNMplot web tool [21]. The web-based tool Gene Expression Profiling Interactive Analysis (GEPIA) was used to explore the expression of *SCN5A* and *SLC9A1* genes according to cancer stage (colon adenocarcinoma, COAD) and to assess the overall survival and disease-free survival (colon adenocarcinoma COAD, rectal adenocarcinoma READ and stomach adenocarcinoma STAD datasets) from the TCGA database [22]. Single-cell *SLC9A1* gene expression was obtained from the dataset experiment “Single cell sequencing of colorectal tumors and adjacent non-malignant colon tissue” deposited in the Single Cell Expression Atlas (EMBL-EBI Expression Atlas) (https://www.ebi.ac.uk/gxa/sc/home; accessed on 29 August 2022) [23]. Tumor and adjacent non-malignant tissues from nine colorectal cancer patients were processed to generate single-cell suspensions which were converted to barcoded 10× libraries of about 7000 cells/library using 10× Genomics technology. Libraries were sequenced on an Illumina NextSeq, HiSeq4000 and NovaSeq. UMAP (Uniform Manifold Approximation and Projection) plots of single-cell visualization of 52,609 cells were generated on the SCEA server clustered by inferred cell types (neighbors: 20). UMAP plots were colored according to the sampling site (normal or tumor), *SLC9A1* expression (heat map, counts per million) and the organism part (left colon, caecum and sigmoid colon).

### 2.5. Culture of Cell Lines

HCT116, HT29, SW480 and SW620 human colon cancer cell lines were purchased from the American Type Culture Collection (LGC Promochem, Molsheim, France). Cancer cells were grown at 37 °C in a 5% CO_2_ incubator, in a humidified atmosphere. Colon cancer cells were cultured in DMEM supplemented with 10% fetal calf serum (FCS). HCT116 shCTL and shNa_V_1.5 as well as SW620 shCTL and shNa_V_1.5 cell lines were generated via transduction with a lentiviral vector encoding a short hairpin RNA (shRNA) specifically targeting human *SCN5A* transcripts (shNa_V_1.5 cell lines) or a null-target shRNA (shCTL cell lines). The sequence encoding sh*SCN5A* was 5′-GCTGGACTTTAGTGTGATTATCTCGAGATAATCACACTAAAGTCCAGC-3′ while the one for a null-target was 5′-CCTAAGGTTAAGTCGCCCTCGCTCGAGCGAGGGCGACTTAACCTTAGG-3′. Cultures were tested weekly for the absence of mycoplasma contamination by MycoAlert™ Mycoplasma Detection Kit (Lonza, Colmar, France).

### 2.6. Transfection of Small-Interfering RNA

SW620 colon cancer cells were transfected with small-interfering RNAs purchased from Tebu-Bio (Tebu-Bio, Le Perray-en-Yvelines, France). Cell monolayers in 6-well plates at 80–90% confluence were incubated with transfection mixes containing 9 µL Lipofectamine RNAi max (Invitrogen) reagent and 20 nM siRNA directed against Na_V_1.5 or 10 nM siRNA against NHE-1 or 20 nM scramble siRNA-A as a control in a final volume of 1 mL using Opti-Mem culture medium (Thermo Fisher Scientific; Waltham, MA, USA). Gene knockdown via transfection was verified by RT-qPCR using the Bio-Rad iCycler Thermal Cycler system (Bio-Rad, Marnes-la-Coquette, France).

### 2.7. Preparation of Colon Tumor-Derived Primary Cell Cultures

Resected biopsies from colorectal tumors were washed, diced and enzymatically dissociated to produce explant cultures according to the conditional reprogramming protocol previously published [24]. Briefly, samples were washed with cooled PBS in a 50 mL falcon tube, mixing by inversion to remove traces of methylene blue, blood and feces. Then, tissues were immersed for 2 s in 90% ethanol and placed immediately after into 10 mL of transport medium (RPMI-1640 supplemented with 2 mM glutamine), penicillin (50 U/mL), streptomycin (50 µg/mL) and amphotericin B (0.5 µg/mL) at 4 °C. Once in the lab, samples were cut into small pieces (<10 mm) and transferred to a 15-mL tube containing 4 mL of F-medium (Composition, Appendix A) supplemented with 5 mg/mL of Collagenase IV and incubated for 1–3 h in agitation and standard conditions (37 °C, 5% CO_2_). After digestion, preparation was centrifuged at 500× *g* for 5 min and the supernatant was discarded. Pellets were then resuspended into 10 mL of fresh F-medium and filtered through a 100-micron cell strainer into a new 50-mL centrifuge tube. Preparations were centrifuged at 300× *g* for 3 min and the supernatants were discarded. Finally, cell pellets were resuspended in F-medium and seeded in 6-well plates coated with collagen I (1 mg/mL). Plates were incubated in standard conditions (37 °C, 5% CO_2_, water-saturated atmosphere). A daily follow-up control was performed to identify successful tissue explants.

### 2.8. Cellular Electrophysiology

Colon cancer cells were plated at a density of 100,000 cells per 35 mm Petri dish. Sodium current recordings were performed between 72 and 120 h after cell seeding. The activity of Na_V_1.5 channels expressed in colon cancer cells was assessed using the patch clamp technique in whole-cell configuration. Recording micropipettes were obtained from borosilicate glass capillaries (TW150−3, World Precision Instruments, Hertfordshire, UK) and a P-97 Flaming/Brown Puller was used (Sutter Instruments, Novato, CA, USA); then, the opening diameter of the micropipette tip was reduced with the help of a micro-forge MF−200 (World Precision Instruments, Hertfordshire, UK) to obtain resistances of 4–6 MΩ. All experiments were conducted at a controlled temperature of 25 °C and sodium currents were recorded using an Axopatch 200B amplifier, a Digidata1322A A/D converter and pCLAMP 8.1 software (Molecular Devices, San Jose, CA, USA). Current digitalization, whole-cell series resistance, cell-capacitance estimations and leakage-current subtraction were performed as previously described [25]. The external solution was composed as follows (in mM): 140 NaCl, 4 KCl, 2 CaCl_2_, 1 MgCl_2_, 11 D-Glucose and 10 HEPES-NaOH (pH 7.4). The internal solution was composed as follows (in mM): 130 KCl, 15 NaCl, 0.37 CaCl_2_, 1 MgCl_2_, 1 Mg-ATP, 10 EGTA, adjusted to pH 7.2 with KOH (1 M). Voltage-activated sodium currents were obtained by applying a rectangular voltage protocol of −5 mV starting at a holding potential (HP) of -100 mV for a duration of 50 ms and at a frequency of 1 Hz. The current–voltage (I–V) relationships of the sodium currents were obtained using a classical I-V protocol from −100 to +60 mV. The steady-state inactivation of sodium channels was explored by applying a classical two-pulse protocol as previously described [25,26,27]. Recovery from the inactivation properties was studied via the application of the protocol described as follows: sodium currents were inactivated using a 50 ms pulse to −5 mV in an initial pre-pulse; then, the membrane potential was stepped to −100 mV for periods ranging from 1 to 400 ms and at that time a 30-ms activating voltage step to −5 mV was applied. Current amplitudes were normalized to cell capacitance and expressed as current density (pA/pF). Activation–voltage relationship curves of sodium channels were built by calculating the conductance through Na^+^ channels (g_Na_) at each test potential by dividing peak current amplitude by the respective driving force (*V*m – *V*rev), where *V*m is the test potential while *V*rev refers to reversal potential. Then, conductance was normalized to its maximal value and plotted against *V*_m_. Data points were then fitted with a Boltzmann function: *G* = *G*_max_/(1 + exp (−(*V*m − *V*_50_)/k)), where *G*_max_ is the maximum Na^+^ conductance; *V*_m_ is the test potential; *V*_50_ is the mid-point of activation and *k* is the slope factor. Experiments to determine the *E*_m_ of colon cancer cell types were performed under a partial rupture of cell membrane in a cell-attached configuration. Briefly, colon cancer cells were perfused with external recording solution and subjected to the cell-attached configuration, exhibiting a high resistance seal (>2 GΩ) under the *Voltage-Clamp* mode of patch amplifier. Then, mode was changed to *Current-Clamp* and the voltage button (*V*_m_) was activated in the meter panel to display millivolt measurements on the panel screen; finally, a partial membrane rupture was performed by applying negative pressure on the cell-attached membrane and the first *E*_m_ value measured on the screen was recorded as an observation of *E*_m_. This type of experiment was repeated on several cells at different passage numbers and then averaged.

### 2.9. Epifluorescence Microscopy

Cancer cells were cultured for 48 h on glass coverslips in standard conditions. Cells were washed with PBS and fixed in 4% of paraformaldehyde (Invitrogen, Thermo Fisher Scientific, Waltham, MA, USA) for 30 min at room temperature. Then, preparations were washed with PBS for 10 min and membrane permeabilization was performed using 100 µM digitonin in PBS for 10 min. Non-specific binding sites were saturated with a blocking solution (5% BSA, 22.52 mg/mL glycine and 0.1% Tween 20 in PBS) for 1 h. Primary antibodies were diluted in 5% BSA solution (anti-Na_V_1.5, 1:300, Alomone Labs, Jerusalem, Israel; ASC-005; anti-NHE-1, 1:100, Santa Cruz Biotechnology, Heidelberg, Germany, sc-58635; anti-Cytokeratin (pan reactive), 1:100, BioLegend, San Diego, CA, USA, BLG628602; anti-Vimentin, 1:100, BioLegend, San Diego, CA, USA, BLG699302; Anti-Sodium Channel Na_V_1.5, 1:100, Sigma-Aldrich, Saint-Quentin-Fallavier, France, S0819; anti-NHE-1, 1:100, Santa Cruz Biotechnology, sc-136239; Anti-Na_V_1.5 Na^+^ CP Vα, 1:100, Santa Cruz Biotechnology sc-271255; anti-NHE-1, 1:100, Abcam, Paris, France, ab67314). Primary antibody solutions were added to the preparations and incubated at 4 °C overnight in a humidified chamber. Three washes of 5 min each were performed and Alexa Fluor 647-coupled secondary-antibody or Alexa Fluor 488-coupled secondary-antibody (1:5000, Thermo Fisher Scientific, Waltham, MA, USA) solutions were added to the preparations and incubated for 1 h at room temperature. Preparations were incubated with DyLight^TM^ 554 Phalloidin (Cell Signaling Technology, Danvers, MA, USA) for 20 min to label the F-actin cytoskeleton. Finally, preparations were mounted with ProLong Gold antifade reagent with DAPI (Thermo Fisher Scientific, Waltham, MA, USA). Negative controls were included for each condition. Epifluorescence images were acquired using the EVOS™ M7000 Imaging System (Thermo Fisher Scientific, Paris, France) using a 60× oil objective (Olympus, Rungis, France). Image analysis was performed using the Fiji ImageJ software 2.9.0/September 2014, National Institutes of Health, Bethesda, MD, USA.

### 2.10. Measurement of Intracellular pH

Cells were treated (pharmacologically or using siRNA) for 24 h before starting the experiment. Cells were incubated for 30 min at 37 °C in Hank’s medium containing 2 µM BCECF-AM (2′,7′-bis-(2-carboxyethyl)-5-(and-6)-carboxyfluorescein; excitation 503/440 nm; emission 530 nm). Excess dye was removed by rinsing the cells twice with physiological saline solution. H^+^ efflux was measured as previously described [14],[15],[16] using the F-2700 Fluorescence Spectrophotometer (Hitachi, Tokyo, Japan) after an NH_4_Cl pulse-wash protocol in the absence of external sodium responsible for intracellular acidification; then, realkalinization occurred after a NaCl extracellular addition to a final concentration of 130 mM.

### 2.11. Two-Dimension Cancer Cell Invasion Assays

Cancer cell invasiveness was assessed using culture inserts with 8-µm pore size migration filters (Becton Dickinson, Pont-de-Clax, France), which we manually covered with a film of Matrigel (300 µg/mL) or collagen I (300 µg/mL). On the upper chamber of the insert, 1 × 10^5^ cells were seeded in 200 µL DMEM with 0.1% FCS and then the lower compartment was filled with 800 µL DMEM supplemented with 10% FCS, thus creating a 100× chemoattractant gradient. Chambers were incubated for 48 h at 37 °C and in 5%-CO_2_. Cells that had invaded were stained with DAPI and visualized using the EVOS™ M7000 Imaging System (Thermo Fisher Scientific, Paris, France). Finally, cells were manually counted on the whole area of the insert membrane. Assays were performed in triplicate for each independent experiment.

### 2.12. Cell Viability

Colon cancer cells were seeded in triplicate in a 96-well plate at a density of 10 × 10^3^ cells per well. Cells were incubated with different concentrations of compound **1** and compound **4** [27] for 48 h; the number of viable cells was assessed by the tetrazolium salt assay (MTT) and normalized to the appropriate control condition (vehicle, 0.1% DMSO).

### 2.13. Chemical, Antibodies and Small-Molecule Na_V_1.5 Inhibitor Compounds

Tetrodotoxin was purchased from Latoxan (Portes-lès-Valence, France). Fluorescent probes and conjugated antibodies were purchased from Fischer Scientific (Illkirch-Graffenstaden, France). All other drugs and chemicals were purchased from Sigma-Aldrich (Saint-Quentin-Fallavier, France). Small-molecule Na_V_1.5 inhibitors were designed using a 3D-QSAR model and obtained using chemical synthesis as previously described [27].

### 2.14. Statistical Analysis

For data presented in violin plots, polygons represent the density estimates and extend to the extreme values, whiskers are extended 1.5 times the interquartile range from the first and third quartiles, white circles show the medians and box boundaries indicate the first and third quartiles. For box plots, the central lines indicate the means, the boxes represent the first and fourth quartiles and the whiskers indicate the minimum and maximum values. For the data presented in the scatter plots, the bars represent the mean values and the error bars represent the associated standard deviation. A one-way ANOVA, followed by the Dunn Multiple Comparison Tests, Kruskal-Wallis test, Mantel-Cox test, Mann–Whitney rank sum tests and paired Student *t*-tests, was used to identify statistical significance. Each experimental result with statistical difference is shown as follows: *, *p* < 0.05; **, *p* < 0.01; ***, *p* < 0.001; ****, *p* < 0.0001, while “ns” means that there was no statistical significance. Data processing for the generation of the graphs and statistical analyses were performed using GraphPad Prism V6 (GraphPad Software, San Diego, CA, USA).

## 3. Results

The aberrant expression and contribution of Na_V_1.5 sodium channels to the aggressive phenotype of leukemia, breast, ovarian and colon cancer cells have been previously described [5,6,7]. In this study, we set out to increase the evidence and explore the joint role of Nav1.5 sodium channels and NHE-1 exchanger in the invasive properties of colon cancer.

### 3.1. Stage- and Anatomy-Dependent Expression of SCN5A Gene in Colon Cancer

In order to explore the expression of the *SCN5A* gene in a large number of clinically relevant samples, we have made use of publicly available transcriptome-level datasets. We performed data mining of the Affymetrix HGU133 Gene Array platform of the NCBI Gene Expression Omnibus which was composed of 377 normal tissues, 1450 tumors and 99 metastatic samples. Results from this cohort show a differential expression of the *SCN5A* gene between groups, being slightly downregulated in the tumor group compared with normal tissue but substantially overexpressed in metastatic samples (Figure 1a). Furthermore, expression levels for the *SCN5A* gene showed an increase with colorectal cancer progression across stages, with the highest level of transcripts in stage IV colon cancer (Figure 1b). Survival analysis was performed using the TCGA dataset. Kaplan–Meier curves were plotted for gastrointestinal cancer patients followed for a duration of 12.5 years (Figure 1c). High *SCN5A* mRNA expression was found to be correlated to worsened overall survival for all cancer patients. A high level of *SCN5A* mRNA expression was also found to be correlated with worse disease-free survival in the same group of cancer patients. We also investigated the expression levels of the gene *SCN5A* using RT-qPCR depending on the tumor site, using colon cancer biopsies surgically collected at the University Hospital of Tours (*n* = 136). We found differential gene expression depending on the anatomical origin of the colon tumors, showing a downregulation from the right colon tumors and upregulation in those that originated on the sigmoid colon epithelium (Figure 1e). Furthermore, *SCN5A* overexpression identified in sigmoid colon tumors was even more pronounced with cancer-stage progression (Figure 1f).

### 3.2. Nav1.5 Is the Main Pore-Forming Alpha-Subunit Isoform Expressed in Colon Cancer Cells

Previous works have demonstrated the expression of pore-forming alpha-subunits of Na_V_ sodium channels in colon cancer [18,19,20,25,28]. Here, we present the relative expression levels for each member of the Na_V_ sodium channel family in colon cancer cell lines assessed using RT-qPCR (Figure 2a). As previously reported, *SCN5A* transcripts encoded for the Na_V_1.5 alpha subunit were the most abundantly expressed in HCT116, HT29, SW480 and SW620 colon cancer cells. It is worth mentioning that the expression levels of the *SCN5A* gene in SW620 cancer cells, a cell line derived from a metastatic tumor, were up to ten times higher than those found in the other cell lines (Figure 2a, blue bars) suggesting the presence of a higher sodium current density in this cell type. However, we also found other Nav-encoding genes being importantly expressed in colon cancer cells, such as *SCN4A* which is significantly expressed in HT29 cells. On the other hand, transcripts from the *SCN8A* alpha subunit-encoding gene were the second most abundant in these cells. Subsequently, we studied the expression of the translated Na_V_1.5 channel protein in colon cancer cells using epifluorescence imaging experiments (Figure 2b). Results showed the expression of the Na_V_1.5 protein at the plasma membrane in the four cell lines studied here. However, the Na_V_1.5 protein signal appeared to be more abundant in SW620 cells with a clear plasma membrane localization (Figure 2b).

### 3.3. Na_V_1.5 Sodium Channels Are Responsible for the Generation of Macroscopic Na^+^ Currents in Colon Cancer Cells

We then investigated and characterized the activity of Na_V_ channels in colon cancer cell lines by measuring macroscopic Na^+^ currents using the patch clamp technique in a whole-cell configuration. Results obtained via electrophysiology experiments showed the presence of voltage-activated sodium currents in the four cell lines studied (Figure 3a). We also studied the steady-state inactivation mechanism (or availability) of Na_V_1.5 sodium channels expressed in these malignant cells. To explore this, we applied inactivating pre-pulses to several *V*_m_ values for 50 ms; then, the fraction of not-inactivated Na_V_1.5 channels was estimated using a test pulse of −5 mV. The families of Na_V_1.5 currents recorded at −5 mV after 50 ms pre-pulses obtained from SW620 cells are shown in Figure 3b. To increase the evidence about the performance of Na_V_1.5 channels in colon cancer cells, recovery from the inactivated state was also explored using a classic two-pulse voltage protocol from a holding potential (HP) of −100 mV (Figure 3b). Sodium currents were inactivated using step depolarization to −5 mV, followed by increasing periods of time at −100 mV, and finally the recovered current was evoked by a second pulse to −5 mV. Representative currents from SW620 cells illustrating the recovery of Na_V_1.5 channels at −100 mV are shown in Figure 3c. We also show the classic inhibitory effect of 30 µM TTX on Na_V_1.5 sodium channels in colon cancer cells (Figure 3d). Next, we built the current–voltage relationships of Na_V_1.5 channels in colon cancer cell lines; to do so, current amplitudes obtained from the *I-V* protocol were normalized to the cell capacitance (*C*_m_) value of each cell, averaged and plotted as a function of the test potential (Figure 3e). In all four cell lines, the Na^+^ current increases steeply from −55 to −5 mV, then it gradually decreases from 0 to +50 mV and becomes outward at more positive voltages (Figure 3e). Results exhibited differences in the magnitudes of current density at −5 mV, with SW620 cells having the highest current density of about −16.8 ± 3.5 pA/pF (*n* = 9), followed by SW480 cells with −6.3 ± 0.4 pA/pF (*n* = 10), HT29 cells with −5.4 ± 0.8 pA/pF (*n* = 14) and HCT116 cells with the lowest current density of −3.3 ± 0.4 pA/pF (*n* = 11) (Figure 3e, Table 1). In order to investigate differences in Na_V_1.5 voltage dependency among colon cancer cells, normalized *I-V* relationships were constructed by relativizing the maximum value of current density as the unit for each cell line, and Boltzmann functions were used to fit the data (Figure 3f). The reversal potential (*V*_rev_) values were +58.2 ± 1.3 mV for HCT116, +49.5 ± 1.0 mV for HT29, +49.8 ± 1.2 mV for SW480 and +54.1 ± 0.9 mV for SW620; the last three, being statistically different from the theoretical equilibrium potential for Na^+^ ions (ENa), were calculated using the Nernst equation (+58.2 mV). The voltage dependence of Na^+^ channel activation was very similar among SW480 and SW620 cells and among HCT116 and HT29 (Figure 3g, Table 1). Sodium conductance was calculated for each cell type by dividing the peak current by the corresponding driving force (*V*_m_-ENa). We then averaged and normalized conductance to generate the activation curves shown in Figure 3g. The voltage dependence of the steady-state inactivation of Na^+^ channels in colon cancer cells was investigated using a classical protocol described above. Results show that the inactivation curves of Na_V_1.5 channels in SW620 cells were shifted significantly towards depolarizing potentials by 8.8 mV compared to HT29 and SW480 curves (Figure 3g, Table 1). Finally, we also investigated the Na_V_1.5 window current in colon cancer cells. The term “window current” refers to the small magnitude of Na^+^ current that can be seen in a range (“window”) of voltages, due to the activation of a proportion of channels which do not inactivate. Here, we studied the window currents appearing in the range of voltages from −60 to −20 mV in colon cancer cells. Our analysis indicates that the intersection point of the curves (channel’s availability and opening likelihood) for each cell line was located at different magnitudes, taking the following order: HCT116 > SW620 > HT29 > SW480 (Figure 3g, inset). We also investigated the membrane potential (*E*_m_) of each cell line and obtained values of −46.2 ± 4.2 mV for HCT116, −27.7 ± 6.4 mV for SW620, −38.5 ± 3.3 mV for HT29 and −42.4 ± 3.8 mV for SW480, with all of them within the range of voltages of window currents (Figure 3g, inset, Table 1). Then, the area under the curve (AUC) was calculated and relativized to SW480 cells obtaining ratios of 1, 1.3, 1.3 and 2 for SW480, SW620, HT29 and HCT116, respectively. Voltage parameters of Na_V_1.5 channels expressed in colon cancer cells are presented in Table 1.

### 3.4. Colon Tumor-Derived Primary Cells Express Functional Na_V_1.5 Sodium Channels

Most of the evidence for the presence of Na_V_ sodium currents in cancer cells has been obtained from established cancer cell lines, although for cervical cancer Nav currents were exceptionally identified in primary cells derived from human tumors [29]. Here, we developed a tissue explant methodology to obtain tumor-derived cells of colon biopsies (Appendix A). Tissue explants were cultured for 2 weeks and cells from the cellular outgrowth were harvested using trypsin to be used in fluorescence imaging, electrophysiology and molecular biology experiments. Figure 4a shows two representative adherent explants surrounded by a typical outgrowth cell monolayer area, then primary cells were harvested and Na_V_1.5 channel proteins were stained with Alexa Fluor 488 (Figure 4a, lower right). An epithelial phenotype of tumor-derived cells was assessed using immunodetection and epifluorescence imaging experiments using a pan-cytokeratin antibody (Appendix A). *SCN5A* transcripts encoding for the Na_V_1.5 alpha subunit were explored both in primary cells and colon cancer cell lines. Results of quantitative PCR showed that the expression levels of *SCN5A* transcripts in primary colon cancer cells were as abundant as those found in SW620 cancer cells (Figure 4b). Subsequently, in order to explore the expression and activity of Na_V_1.5 channel proteins, we made whole-cell electrophysiological recordings from these colon tumor-derived primary cells. We found inward sodium currents activated with a voltage step at −5 mV from an HP of −100 mV which were blocked by the addition of 30 µM TTX (Figure 4c). We also explored voltage-dependence activation of Na_V_1.5 channels expressed in primary cells using a classical *I-V* voltage protocol. The families of Na_V_1.5 current recordings of these cells showed step increments from −55 to 0 mV, then Na^+^ currents gradually decreased from 0 to +60 mV. The averaged current density at −5 mV was −2.5 ± 0.9 pA/pF (*n* = 6) and the reversal potential value was found at +63.6 ± 5.1 mV (Figure 4d, e). Interestingly, a voltage-dependent outward current component activated around -20 mV was identified (Figure 4d). Recording solutions do not contain potassium channel blockers; therefore, such an outward current could be due to some K_V_ channel expressed in these cells. We also studied the steady-state inactivation mechanism and calculated the normalized conductance as we mentioned above, and we obtained a *V*_50_-inactivation of −49.4 ± 3.7 mV, a *V*_50_-activation of −14.3 ± 0.6 mV and found that the area under the overlap of curves was 2.3 times higher compared to SW480 cells (Figure 4f, inset, Table 1).

### 3.5. SLC9A1 Gene, Encoding for the Na^+^/H^+^ Exchanger Type 1, Is Differentially Expressed Regarding Anatomical Location of Colon Tumors

It has been demonstrated that Na_V_1.5 channels enhance the invasiveness of breast cancer cells by acting as an allosteric modulator of the Na^+^/H^+^ exchanger type 1 (NHE-1) in invadopodial structures, thus promoting H^+^ efflux and the activation of acidic-dependent proteases [14,15,16]. We explored the expression levels of the *SLC9A1* gene in the same publicly available transcriptome datasets mentioned above (377 normal tissues, 1450 tumors and 99 metastatic samples). Surprisingly, results showed a downregulation of the *SLC9A1* gene in tumor and metastatic groups compared to the non-cancer tissue, (Figure 5a). However, no significant change was found in *SLC9A1* expression levels over the course of colon cancer progression (Figure 5b). On the other hand, analysis via Kaplan–Meier curves indicates that those cases of colon cancer with high levels of *SLC9A1* transcripts are correlated with worsened overall survival, but are without correlation with disease-free survival data (Appendix A). In order to explore in detail the expression of the *SLC9A1* gene in colon cancer we also used a single-cell RNA-seq dataset identified as “Single cell sequencing of colorectal tumors and adjacent non-malignant colon tissue” deposited in the EMBL-EBI Expression Atlas (www.ebi.ac.uk/gxa; accessed on 29 August 2022). Uniform Manifold Approximation and Projection (UMAP) plots show single-cell visualization of 52,609 cells from tumors and adjacent non-malignant tissues of nine patients clustered according to inferred cell types (neighbors: 20). UMAP plots were first colored according to sampling site, highlighting all cell types from normal tissue (orange) and those cells from the tumor core (blue) (Figure 5d). Then, we assessed the expression levels of the *SLC9A1* gene in the same cell cluster distributions, showing its expression levels in counts per million and plotted as a heat map within the UMAP plot (Figure 5e). Results showed a heterogeneous distribution of the expression of *SLC9A1* transcripts that were even almost absent for some cellular clusters (Figure 5e). Then, clusters were colored according to the anatomical location of the colon from which the samples were taken, displaying left colon (brown), caecum (light green) and sigmoid (dark green). Results showed that cells from colon cancer tumors originating from the epithelium of the caecum and sigmoid are mainly enriched with *SLC9A1* transcripts (Figure 5f). We also investigated the expression levels of the *SLC9A1* gene in colon cancer biopsies collected at the University Hospital of Tours (*n* = 136) and found a significant overexpression in tumors originating from the caecum, right and sigmoid colon (Figure 5g) in accordance with the single-cell RNAseq data. Additionally, we made use of the immunohistochemical evidence for the NHE-1 protein in normal and tumor colon tissue deposited in The Human Protein Atlas consortium (www.proteinatlas.org; accessed on 5 May 2022). We identified an overexpression of NHE-1 in tumor samples, in which tumor cells showed high-intensity protein staining with a plasma membrane and cytoplasmic signal localization (Figure 5h).

### 3.6. Na^+^/H^+^ Exchanger Type 1 Contributes to Colon Cancer Cell Invasiveness

It has been demonstrated that several members of the Na^+^/H^+^ exchanger family are expressed in colon cells and contribute to pH regulation [30]. Here, we investigated the expression levels of four plasma membrane Na^+^/H^+^ exchanger isoforms (NHE1-4, encoded by *SLC9A1*-4 genes, respectively) in HCT116, HT29, SW480 and SW620 colon cancer cells. Results showed that the *SLC9A1* isoform was the most expressed in all cell lines followed by *SLC9A2*, *SLC9A3* and finally *SLC9A4*. *SLC9A1* transcripts were found to be in the same expression levels among cell lines with no significant difference (Figure 6a). The expression of NHE-1 proteins was explored using epifluorescence microscopy of colon cancer cells. Results showed the expression of the NHE-1 protein at the plasma membrane but a cytoplasmic staining was also found in the four cell lines studied here (Figure 6b). Proton effluxes were assessed in SW620 cells that were acidified via an incubation with NH_4_Cl and then resuspended in a sodium-free solution. H^+^ efflux was triggered by the addition of 130 mM NaCl and monitored using fluorescent-based measuring of intracellular pH. The recovery of intracellular pH after the addition of NaCl under control conditions (light blue) or in the presence of the NHE-inhibitor EIPA (dark blue) is shown in Figure 6c. EIPA reduced H^+^ efflux by about 50 and 90%, respectively. Then, in order to find out the specific contribution of NHE-1 to proton efflux, we transfected SW620 cells with small interfering RNAs against *SLC9A1* transcripts (*siSLC9A1*) compared to the transfection of irrelevant siRNA (*siCTL*) and assessed the relative H^+^ efflux. Results showed that NHE-1 is responsible for about 40% of the total H^+^ efflux while the other NHE family members possibly contribute to the remaining 50% of proton exchange, which was inhibited by the addition of EIPA (Figure 6d). However, about 20% of the total H^+^ efflux was not dependent on NHE exchangers, since they were not affected by pharmacologic or transcriptional inhibition of NHE proteins (Figure 6d). Next, we evaluated the modification of relative H^+^ efflux using TTX, EIPA or the combined treatment. Independent conditions of cells transfected with siRNA targeting *SCN5A* transcripts in the absence or presence of EIPA were also studied. The results showed that TTX, EIPA and the combination of the two decreased about 60% of H^+^ efflux, whereas silencing Na_V_1.5 channels did so by 30% alone, although in combination with EIPA the decrease was up to 70% (Figure 6e). Finally, we assessed the 2D invasiveness of colon cancer cells under the pharmacological inhibition of Na_V_1.5 channels using TTX, of NHE exchangers using EIPA, in cells transfected with irrelevant small interfering RNA (siCTL), or targeted to *SCN5A* (si*SCN5A*) or *SLC9A1* (si*SLC9A1*). The results show that inhibiting Na_V_1.5 showed no additive effect to the inhibition of NHE-1, suggesting a common signaling pathway. In addition, NHE-1 silencing decreased invasiveness by approximately 70%, with no further decreases occurring in combination with TTX or EIPA (Figure 6f).

### 3.7. Na_V_1.5 Sodium Channels and NHE-1 Exchanger Proteins Colocalize in Colon Cancer Cells

Previous evidence has shown a regulation of the internal and perimembrane pH via the inhibition of Na_V_1.5 activity with TTX [14], and shortly thereafter a colocalization of Na_V_1.5 sodium channels and NHE-1 exchanger in regions rich in Caveolin-1 at the plasma membrane of breast cancer cells was found [15]. Furthermore, these proteins were not only identified to colocalize at the plasma membrane of cancer cells but their direct interaction was also demonstrated by proximity ligation assays at the highly specialized invadopodial membrane structures, which together with the proteases cathepsin-B and membrane-type 1 matrix metalloproteinase (MT1-MMP) generate sites of degradation of extracellular matrix [16]. Here, we studied the colocalization of Na_V_1.5 sodium channels and NHE-1 exchanger in colon cancer cells via epifluorescence microscopy using two sets of primary antibodies (Appendix A) against these two proteins (Figure 7, Appendix A). The results obtained with the first set of primary antibodies (Anti-Sodium Channel Na_V_1.5 antibody produced in rabbit Sigma-Aldrich Ref. S0819; anti-NHE-1 produced in mouse Santa Cruz Biotechnologies Ref. sc-136239) showed that both proteins are mainly localized in the plasma membrane of SW620 colon cancer cells (Figure 7a). Subsequent analysis with merged images indicate that the corresponding fluorescence signals for each protein overlap in the bidimensional space (Figure 7b), with a correlation coefficient of 0.957 obtained using cytofluorogram analysis (Figure 7c) and a Pearson Coefficient r = 0.97 obtained using Costes’ automatic threshold analysis (Figure 7d). Similar results were obtained with the second set of primary antibodies used in this study (Appendix A). In addition, we have performed a protein–protein interaction prediction analysis based on the amino acid sequence of Na_V_1.5 (Uniprot ID: Q14524-1) and NHE-1 (Uniprot ID: P19634-1) proteins using the online tool Protein–protein Affinity Predictor (PPA-Pred2, https://www.iitm.ac.in/bioinfo/PPA_Pred/index.html; accessed on 5 December 2022) [31]. Results showed a predicted value of binding free energy, ΔG = −4.10 kcal/mol, and a dissociation constant, Kd = 9.86 × 10^−04^ M (Appendix A), which suggests that the interaction between the two proteins has a low but real probability of occurring considering that they tend to colocalize and enrich their expression in subcellular structures as described above. Additionally, through the database Search Tool for Recurring Instances of Neighbouring Genes (STRING, https://string-db.org/cgi/input?sessionId=bWoctneOwNlx&input_page_show_search=on; accessed on 5 December 2022) [32] we have identified a network of protein–protein interactions involving Na_V_1.5 sodium channels and the NHE-1 exchanger in which both are co-expressed in different systems and exhibit a direct interaction with calmodulin proteins (Appendix A). Moreover, through *SCN5A* gene silencing experiments using siRNA or short hairpin RNA in stable colon cancer cell lines, we identified that the *SLC9A1* gene is also significantly downregulated (Figure 7e,f), suggesting that the relationship between these proteins is not only through a physical interaction with a pro-invasive role in cancer cells, but probably also through mechanisms of co-regulation of transcript expression.

### 3.8. New Small-Molecule Na_V_1.5 Inhibitors with Potential Anti-Metastatic Activity

Pharmacological inhibition of Na_V_1.5 channels by drugs used clinically to treat heart disease or epilepsy has been shown to decrease the invasive properties of cancer cells in both preclinical and in vitro models [8,9]. Recently, assisted by a 3D-QSAR model, we have designed, synthesized and evaluated new small-molecule drugs with inhibitory activity on Na_V_1.5 channels expressed in breast cancer cells [27]. Here, we characterized the effects of two of these molecules on the biophysical properties of sodium currents expressed in colon cancer cells, as well as their anti-invasive properties on SW620 and HCT116 cells in 2D invasion models. In order to identify the IC_50_ of each compound on Na_V_1.5 channels expressed in SW620 colon cancer cells, we tested the effect of increasing concentrations of the compounds named compound **1** and compound **4** on the peak Na^+^ currents evoked by step depolarization at −5 mV from an HP of −100 mV applied every 2 s. An example of such experiments for compound **1** is illustrated in Figure 8a. We also analyzed the time course of Na^+^ current blocking and results indicate that the inhibitions of Na^+^ currents at each drug concentration are established on average after 60–70 depolarizing steps (Figure 8b). Similar experiments were performed with compound **4** (Figure 8c and Appendix A). The experimental data of Na_V_1.5 current inhibition for compounds **1** and **4** were fitted with Hill’s function obtaining IC_50_ of 2.1 µM and 1.9 µM, respectively (Figure 8d). We next investigated the voltage dependence of Na_V_1.5 inhibition using these two small-molecule drugs. Using the *I-V* voltage protocol described above, the families of Na^+^ currents were recorded in the absence or presence of compound **1** or compound **4**. *I-V* relationships were generated for each experimental condition (Figure 8e). Sodium conductance was calculated in the absence and presence of compound **1** or compound **4** by dividing peak current by the corresponding driving force. We then averaged and normalized conductance to generate the activation–voltage relationships shown in Figure 8f. The voltage dependence of the steady-state inactivation of Na^+^ channels under these conditions was investigated using a classical protocol as described above. Results showed that both activation– and inactivation–voltage relationships of SW620 Na_V_1.5 channels were significantly shifted towards hyperpolarizing potentials in the presence of compound **1** and compound **4** (Figure 8f, Table 2) with a decrease in the area under the curve (AUC) for the window current of 6% and 30% for compound **1** and compound **4**, respectively (Figure 8f, inset). Then, in order to obtain the toxicity of the small-molecule drugs studied here, we performed cell viability assays on SW620 and HCT116 cells. Colon cancer cells were incubated with different concentrations of either compound **1** or compound **4** for a duration of 48 h. Cell viability was estimated via MTT assay using a control condition containing 0.1% DMSO as vehicle. The experimental data were fitted with a Hill function and the IC_50_ values obtained are for SW620 cells of 22 and 6 µM for compound **1** and compound **4**, respectively, and for HCT116 cells of 20 and 6 µM for compound **1** and compound **4**, respectively (Figure 8g, Table 2). Finally, in order to investigate the anti-invasive activity of these two small-molecule drugs on colon cancer cells, we performed 2D invasion assays using Matrigel-coated cell culture inserts on SW620 and HCT116 cells stably expressing a short hairpin RNA targeting *SCN5A* gene expression (shNa_V_1.5) or a null-target shRNA (shCTL). The results showed that compound **1** and compound **4**, both used at the non-cytotoxic concentration of 1 µM, reduced the invasion ability of SW620 cells by 30% and 50%, respectively (Figure 8h). *SCN5A* silencing reduced the invasiveness of SW620 by 50% and neither compound **1** nor compound **4** caused a further decrease, suggesting that these compounds reduce cancer cell invasiveness by selectively targeting Na_V_1.5. TTX (30 µM) was used as a comparative control and induced a similar invasiveness inhibition to the compounds tested. For HCT116 cells, compound **1** and compound **4** inhibited the invasiveness of the cells by 45% and 55%, respectively. *SCN5A* silencing also decreased the invasiveness of HCT116 cells by 50%. Again, small-molecule drugs did not further decrease this invasive property (Figure 8h). The raw number of invasive colon cancer cells in 2D invasion experiments under these conditions is presented in Appendix A.

## 4. Discussion

It has been more than thirty years since the first evidence was found of the presence of voltage-dependent sodium currents in cells derived from myelogenous leukemia [33]; shortly after, it was discovered that sodium channels contribute to the invasiveness of cancer cells [34]. Since then, a large body of evidence has accumulated describing the involvement of the alpha and beta subunits of the Na_V_ sodium channel family in the metastatic behavior of several carcinomas [5,6,7]. The mechanistic aspects of the contribution of Na_V_ channels to the invasive properties of cancer cells are not fully elucidated. However, most of the evidence to date has been obtained using in vitro and preclinical models of breast cancer [8,9,14,15,16,26,35]. A pioneer study showed for the first time the involvement of Na_V_ channels and the importance of the “window current” in the invasive properties using the highly metastatic MDA-MB-231 breast cancer cells [35]. This biophysical property is especially relevant in cancer cells for which the resting membrane potential is generally depolarized to values in the range on that window of voltage [13,35,36].

In this work, we found that the *SCN5A* expression levels increase in the different stages of colon cancer as the tumor progresses. Our analysis also shows that *SCN5A* transcripts are down-regulated for some cases depending on the anatomical location where tumors develop. However, as the disease worsens and metastases appear, there is a strong overexpression of the *SCN5A* gene. This has been observed for other alpha subunits of Nav channels in other cancer types [17]. The prognostic value of Na_V_1.5 channel expression for colon cancer was assessed in patients with tumors in stage I-III, showing that high Na_V_1.5 expression is associated with an unfavorable oncologic prognosis [37]. Our analysis of public databases shows that those cases of cancers of the gastrointestinal tract with high levels of *SCN5A* transcripts are associated with a poor prognosis of both overall survival and disease-free survival. We have also found that the *SCN4A* and *SCN8A* subunits are expressed at significant levels, but their role in colon cancer cell properties such as invasive capacities are not yet identified. It is well known that the product generated by alternative splicing of the *SCN5A* gene known as the neonatal variant, nNa_V_1.5, is also expressed and associated with breast cancer progression [38,39]. Both the neonatal and adult variant are expressed in colon cancer cells as well [28], and their inhibition by the local anesthetic ropivacaine decreased cell invasiveness by approximately 30% when half of the Na^+^ current was inhibited [19]. The neonatal variant of Na_V_1.5 sodium channels differs from the adult isoform in seven amino acids located in S3 and S4 of domain-I. This change generates modifications in the biophysical properties of the nNa_V_1.5 channel protein in comparison to the adult; for instance, the nNa_V_1.5 sodium channels exhibit much slower activation and inactivation kinetics, show a significantly (~50%) greater Na^+^ influx and show a depolarized value for *V*_50_ activation, among others [40]. However, in this work we focused on exploring only the *SCN5A* adult isoform at the transcriptional level by using specific qPCR primers against this variant. Nevertheless, the antibodies used to detect Na_V_1.5 sodium channels are able to recognize both splicing variants. Thus, additional experiments are needed to dissect the role and contribution of each Na_V_1.5 variant in this type of cancer.

We also found differences in terms of Na^+^ current density and window current properties among the cell lines studied here, with the window current perhaps being the most relevant because colon cancer cells exhibit a less hyperpolarized resting potential in comparison to excitable cells, as it has been identified in other types of carcinomas [13,35,36,39,41]. This work demonstrates for the first time the activity of voltage-gated sodium channels in colon tumor-derived primary cells, which exhibits a characteristic voltage dependence and a significant window current as well. Notably, a voltage-dependent outward component similar to that generated by a delayed rectifier K^+^ channel was also identified in these cells. However, we did not explore in detail this type of ionic current, the channel responsible for generating it nor its possible role in these cells.

We identified a correlation between a bad survival prognosis in colon cancer patients and high expression levels of the *SLC9A1* gene, similar to that found in gastric cancer patients [42]. However, the apparent discrepancy seen through the absence of change in *SLC9A1* expression levels throughout the stages of colon cancer, as well as its downregulation in the metastatic group of patients, might suggest that protein activity rather than transcript expression is the most relevant for conferring aggressiveness to colon tumors. NHE-1 together with bicarbonate anion transporters are ubiquitously expressed and are important modulators of intracellular pH [43]. In addition, NHE-1 activity is enhanced by Na_V_ channels in breast and gastric cancer cells [15,42]. Sodium homeostasis is altered in the tumor microenvironment and intracellular Na^+^ concentration found in cancer cells is higher than in normal cells [12,44,45]. The expression and activity of Na^+^ transporter proteins, including Na_V_ channels, are responsible for the modification of sodium balance in cancer cells. It is well known that the Na^+^/Ca^2+^ exchanger type 1 (NCX1), which generally transports Ca^2+^ out in exchange for Na^+^ ions, operates predominantly in reverse or “retrograde” mode in cancer cells [46,47,48]; consequently, intracellular Ca^2+^ increases the promoting of the formation of a Ca^2+^-Calmodulin complex which regulates the functioning of NHE-1 [49].

We found that NHE-1 and Na_V_ sodium channels colocalize at the plasma membrane of colon cancer cells. Then, by amino acid sequence-based tools for predicting protein-protein interactions we identified a direct interaction of NHE-1 and Na_V_1.5 proteins as this has been experimentally identified in invadopodial structures of breast cancer cells, where both proteins promote the extracellular matrix degradation [16]. Here, our analysis provides evidence of the expression of several members of the NHE family and, for the first time, we describe an anatomo-specific dysregulation of the *SLC9A1* gene in human colon tissue. By using transcriptional data obtained using single-cell RNAseq we identified a preferential overexpression of the *SLC9A1* gene in tumors derived from the sigmoid colon and caecum. NHE-1 was the most abundant and the main regulator of H^+^ efflux in colon cancer cells. The activity of NHE-1 and Na_V_1.5 channels contribute to the invasive properties of colon cancer. Mechanistic aspects of the role of Na_V_1.5 channels in colon cancer have been previously addressed by the exploration of in vitro models, human biopsies and bioinformatics analyses demonstrating that these channels are key regulators of a transcriptional network of genes controlling invasion, which includes MAPK signaling via ERK phosphorylation and the PKA/ERK/c-JUN/ELK-1/ETS-1 transcriptional pathway [18,20]. Recently, it has been shown that pharmacological and transcriptional inhibition of rho-associated protein kinase (ROCK) increases the expression of Na_V_1.5 channels at the plasma membrane of colon cancer cells, boosting their invasive capacity [25]. Finally, we also studied the efficacy of two new small-molecule inhibitors against Na_V_1.5 channels. These drugs decreased cell invasiveness as much as other antagonists previously studied in colon cancer and other types of cancer [8,9,19,27]. Thus, such molecules could be proposed as models for the development of new anti-metastatic drugs.

## 5. Conclusions

The aberrant expression of several pore-forming alpha-subunits of Na_V_ channels in tumors of different organs is now well established. This work demonstrates that the dysregulation of Nav channel expression is associated with that of the NHE-1 exchanger and is correlated with increased invasive properties of colon cancer cells. Furthermore, this may occur in an anatomically specific manner or even at the level of cell subpopulations within the tumor. Understanding the mechanisms of Na_V_ channel dysregulation at the cellular subpopulation level will allow these proteins to be used as molecular targets to prevent metastases; this may be particularly suitable for colon cancer because of the well-established procedures of endoscopic intervention that could easily adapt the localized administration of inhibitors of these proteins.

## Figures and Tables

**Figure 1 cancers-15-00046-f001:**
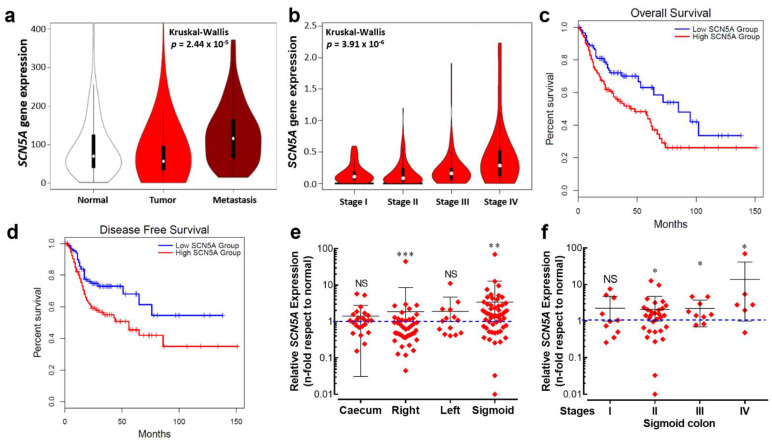
An aberrant expression of the *SCN5A* gene is associated with poor prognosis for cancer patients. (**a**) Violin plots of *SCN5A* gene expression in colorectal tissues from normal (*n* = 377), tumor (*n* = 1450) and metastatic (*n* = 99) specimens showed significant differences (Kruskal–Wallis, *p* = 2.44 × 10^−5^). Data were obtained from the Affymetrix HGU133 Gene Array platform of the NCBI Gene Expression Omnibus repository and processed using the TNMplot tool. (**b**) Violin plots of *SCN5A* gene expression based on pathological colorectal cancer stages using the dataset of The Cancer Genome Atlas (TCGA) processed with the GEPIA visualization tool, Kruskal–Wallis, *p* = 2.44 × 10^−6^. (**c**,**d**) Kaplan–Meier overall survival and disease-free survival for gastrointestinal tract cancer patients. The low *SCN5A* expression (blue lines) and high *SCN5A* expression (red lines) groups were compared using the two-sided log-rank test which obtained significant results for overall survival (HR 1.7, *p* = 0.0037) and disease-free survival (HR 1.8, *p* = 0.0047). (**e**) Relative *SCN5A* gene expression in tumors from the main anatomical structures of the colon. Tumor biopsies from caecum (*n* = 24), right (*n* = 43), left (*n* = 14) and sigmoid (*n* = 55) colon collected at the Hospital of Tours were analyzed and compared with their corresponding adjacent normal tissue. Scatter dot plot shows the average fold-change values (2^−ΔΔCt^) of the *SCN5A* gene for individual samples of each group. (**f**) Scatter dot plot showing *SCN5A* gene expression based on pathological cancer stages of colon sigmoid tumors. * *p* < 0.05; ** *p* < 0.01; *** *p* < 0.001; Mann–Whitney rank sum test.

**Figure 2 cancers-15-00046-f002:**
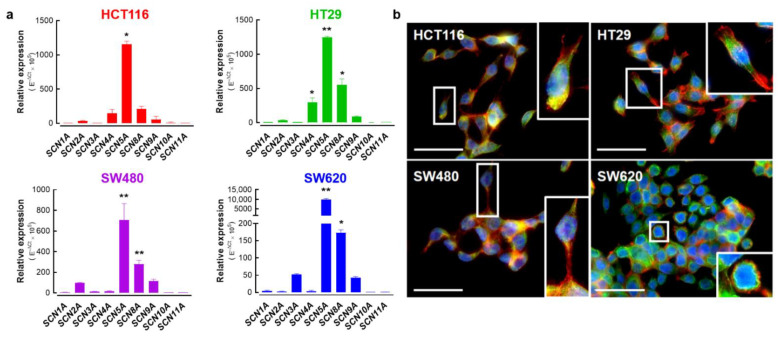
Expression of alpha-subunit voltage-gated sodium channels in colorectal cancer cell lines. (**a**) Bars represent the relative expression (E^−ΔCt^ × 10^5^) of genes encoding for alpha subunits of voltage-gated sodium channels (*SCN1A*-*SCN11A*) in HCT116, HT29, SW480 and SW620 colorectal cancer cells. The *SCN5A* gene, followed by *SCN8A,* were the most expressed genes in these cell lines. *SCN4A* was also importantly expressed in HT29 cells. (**b**) Epifluorescence microscopy analysis of Na_V_1.5 proteins in HCT116, HT29, SW480 and SW620 cells. Images show the staining for Na_V_1.5 proteins (green, Alexa Fluor 488), F-actin filaments (red, phalloidin-594) and nuclei (blue, DAPI). Na_V_1.5 is mainly located at the plasma membrane of cancer cells. For better visualization, a threefold digital amplification was performed for the cell framed within the white box and shown on the right side of each image. Scale bar 50 µm. * *p* < 0.05; ** *p* < 0.01; Mann–Whitney rank sum test.

**Figure 3 cancers-15-00046-f003:**
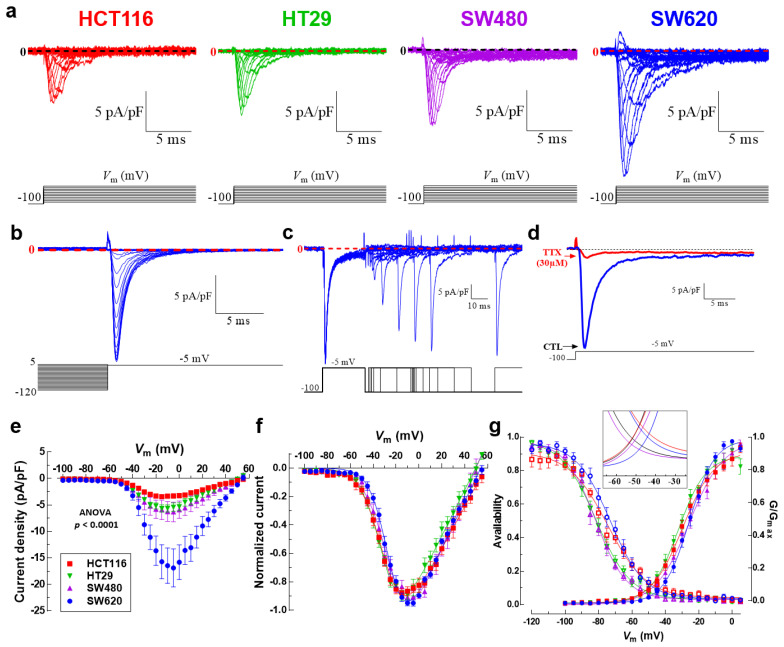
Voltage-gated sodium channel activity in colorectal cancer cells. (**a**) Family of Na_V_1.5 sodium current recordings obtained in response to the illustrated voltage protocol. Macroscopic sodium currents in HCT116, HT29, SW480 and SW620 cells were recorded using the patch clamp technique in the whole cell configuration and evoked by applying a classical I-V voltage protocol and using a holding potential (HP) of −100 mV. (**b**) Na_V_1.5 channels steady-state inactivation in SW620 cancer cells. Families of Na_V_1.5 currents at −5 mV after 50 ms pre-pulses to the potential between −120 and 10 mV. For the purpose of clear presentation, only the last 5 ms are shown. (**c**) Recovery from the inactivation at −100 mV of Na_V_1.5 sodium channels expressed in SW620 cells. A representation of the two-pulse protocol used is shown at the bottom. Na^+^ currents were inactivated by a 50 ms pulse to −5 mV; then, the membrane potential was stepped to −100 mV for periods ranging from 1 to 400 ms and at that time a 30 ms activating voltage step to −5 mV was applied. (**d**) Na^+^ currents of SW620 evoked at −5 mV from an HP of −100 mV under control condition (blue trace) or after the stationary Na_V_1.5 blocking by 30 µM of TTX (red trace) are shown. (**e**) Current–voltage relationships of Na_V_1.5 channels obtained from HCT116, HT29, SW480 and SW620 cells. Peak current amplitudes were normalized to the membrane capacitance (*C*_m_) value of each cell. Data points are averages of 9–14 cells. (**f**) Normalized I-V curves for the same cells shown in (**e**). Solid lines in (**f**) show the fits to the data obtained using a Boltzmann function. The corresponding parameters are summarized in Table 1. (**g**) Activation (filled symbols) and steady-state inactivation–voltage relationships (empty symbols) of Na_V_1.5 channels from HCT116, HT29, SW480 and SW620 cells obtained under standard recording conditions. Smooth lines fit a Boltzmann function and *V*_50_ values were calculated for each cell line. *V*_50_-activation–voltage values obtained were as follows: HCT116 −29.6 ± 0.7 mV; HT29 −32.1 ± 0.7 mV; SW480 −25.7 6 ± 1.1 mV and SW620 −25.6 ± 0.5 mV, while the *V*_50_-inactivation–voltage values corresponded to HCT116 −74.6 ± 0.9 mV; HT29 −81.5 ± 0.8 mV; SW480 −81.7 ± 0.6 mV and SW620 −72.9 ± 0.8 mV. Data are from 9 to 14 cells for each observation. The inset magnifies the individual window currents identified for HCT116, HT29, SW480 and SW620 cells under standard recording conditions.

**Figure 4 cancers-15-00046-f004:**
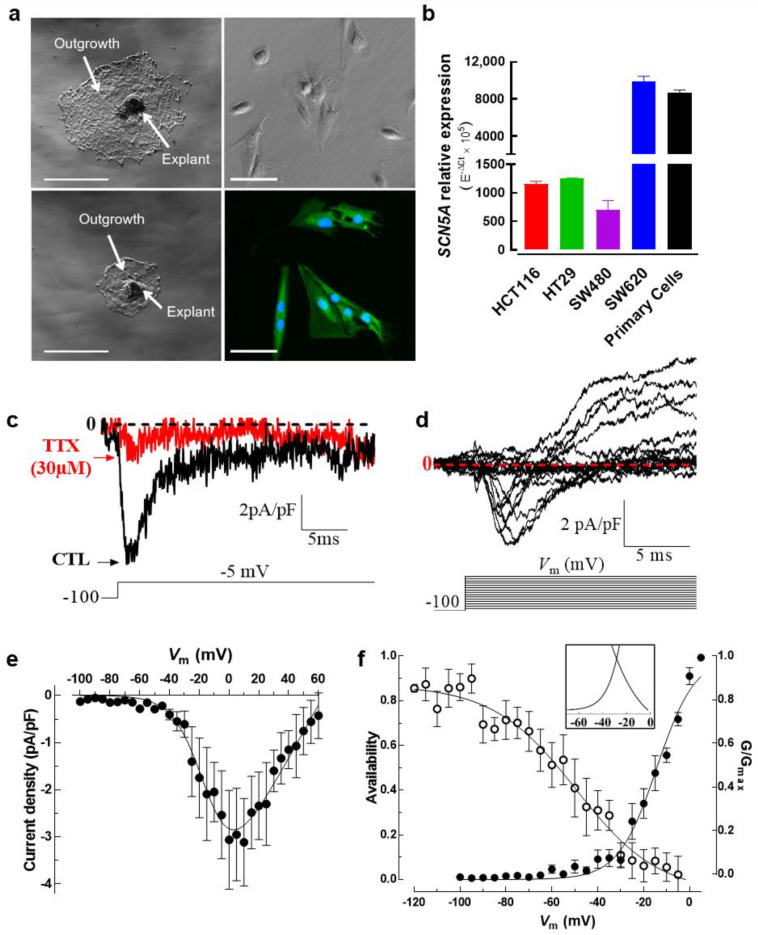
Establishment of tumor-derived primary cells from adherent explants of human colorectal tissue. (**a**) Resected endoscopic biopsies from colorectal tumors were washed, diced and enzymatically dissociated to produce explant cultures. Preparations from digested tissue were seeded in 6-well plates coated with collagen I (1 mg/mL) and incubated under standard conditions (37 °C, 5% CO_2_, water-saturated atmosphere). A daily follow-up control was performed to identify successful explants of colorectal tumors. Low magnification images using phase contrast microscopy showing two adherent explants surrounded by an outgrowth (middle panel section). Scale bar 300 µm. Phase contrast (upper right) and epifluorescence (lower right) microscopy images of tumor-derived primary cells obtained from explant cultures. Fluorescent image shows the staining for Na_V_1.5 protein (green, Alexa Fluor 488) and nuclei (blue, DAPI). (**b**) Bars represent the relative expression (E^−ΔCt^ × 10^5^) of the *SCN5A* gene in colon tumor-derived primary cells compared to colorectal cancer cell lines assessed using RT-qPCR. Primary cells show *SCN5A* mRNA abundance levels similar to SW620 cells (*n* = 5). (**c**) Patch clamp recordings in colon tumor-derived primary cells show Na^+^ currents evoked at −5 mV from an HP of −100 mV under control condition (black trace) which were inhibited by 30 µM TTX (red trace). (**d**) Family of Na_V_1.5 sodium current recordings obtained in response to the illustrated voltage protocol and using an HP of −100 mV. The expression of macroscopic sodium currents in colon tumor-derived primary cells was explored by applying a classical I-V voltage protocol. (**e**) Current–voltage relationships of Na_V_1.5 channels obtained from the cells that were explored in (d). Peak current amplitudes were normalized to the membrane capacitance (*C*_m_) value of each cell (current density in pA/pF). (**f**) Activation (filled symbols) and steady-state inactivation–voltage relationships (empty symbols) of Na_V_1.5 channels from colon tumor-derived primary cells obtained under standard recording conditions. Smooth lines fit a Boltzmann function and *V*_50_ values were calculated. *V*_50_-activation–voltage: −14.2 ± 0.6 mV; *V*_50_-inactivation–voltage: −49.5 ± 3.7 mV (*n* = 7). The inset shows the window current identified for primary cells that was 2.3 higher than that of colon cancer cell line SW480.

**Figure 5 cancers-15-00046-f005:**
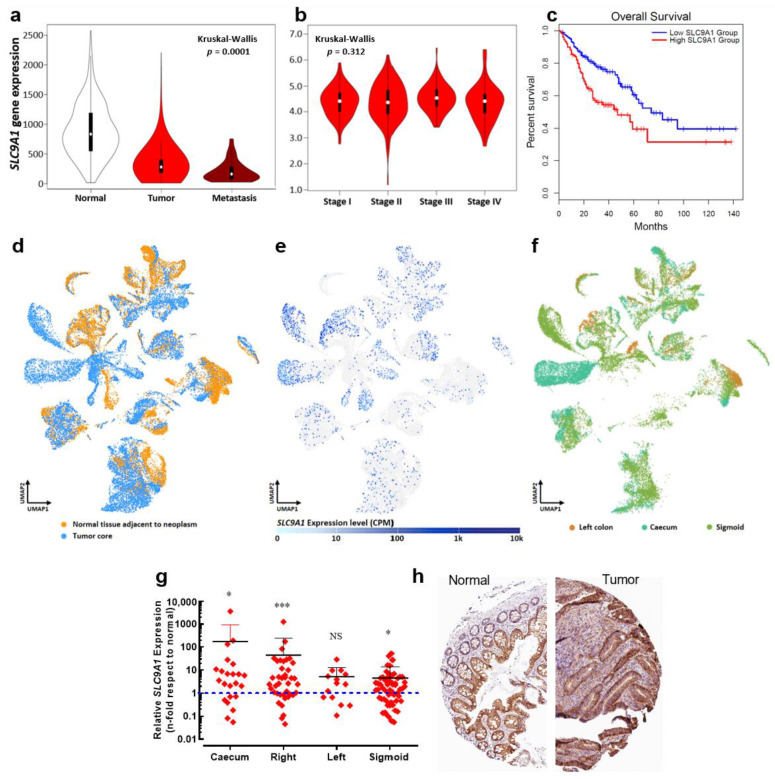
The *SLC9A1* gene is differentially expressed regarding to the anatomical location of colon tumors. (**a**) Violin plots of *SLC9A1* gene expression in colorectal tissues from normal (*n* = 377), tumor (*n* = 1450) and metastatic (*n* = 99) specimens showed significant differences among groups (Kruskal–Wallis, *p* = 1.06 × 10^−12^). Data were obtained from the Affymetrix HGU133 Gene Array platform of the NCBI Gene Expression Omnibus repository and processed using the TNMplot tool. (**b**) Violin plots of *SLC9A1* gene expression based on pathological colorectal cancer stages using the dataset of The Cancer Genome Atlas (TCGA) and processed using the GEPIA visualization tool. (**c**) Kaplan–Meier overall survival for gastrointestinal tract cancer patients. The low *SLC9A1* expression (blue lines) and high *SLC9A1* expression (red lines) groups were compared using the two-sided log-rank test obtaining significant differences (HR 1.9, *p* = 0.00086). (**d**) Data of single-cell RNA-seq of colorectal cancer experiments were obtained from the EMBL-EBI Expression Atlas database (www.ebi.ac.uk/gxa; accessed on 29 August 2022) using the experiment “Single cell sequencing of colorectal tumors and adjacent non-malignant colon tissue”. Briefly, tumor and adjacent non-malignant tissues from colorectal cancer patients were processed and used for single-cell RNA-seq with 10× Genomics. UMAP plots show single-cell visualization of 52,609 cells from nine colorectal cancer patients clustered according to inferred cell types (neighbors: 20). For clarity purposes, the plot colors cells according to their sampling site (normal or tumor tissue). (**e**) Expression levels in counts per million (CPM) of the *SLC9A1* gene across 52,609 cells, as illustrated in (**d**). (**f**) UMAP plot shows colored cells according to the anatomical location from which they originate: left colon, caecum or sigmoid. (**g**) Relative *SLC9A1* gene expression in tumors from the main anatomical structures of the colon. Tumor biopsies from the caecum (*n* = 23), right (*n* = 40), left (*n* = 14) and sigmoid (*n* = 58) colon were analyzed and compared with their corresponding adjacent normal tissue. Scatter dot plot shows the average fold-change values (2^−ΔΔCt^) of the *SLC9A1* gene for individual samples of each group * *p* < 0.05; *** *p* < 0.001, Mann–Whitney rank sum test. (**h**) Immunohistochemical images from the Human Protein Atlas (https://www.proteinatlas.org; accessed on 5 May 2022) showing NHE-1 (*SLC9A1* gene) protein expression in normal and colorectal cancer tissue.

**Figure 6 cancers-15-00046-f006:**
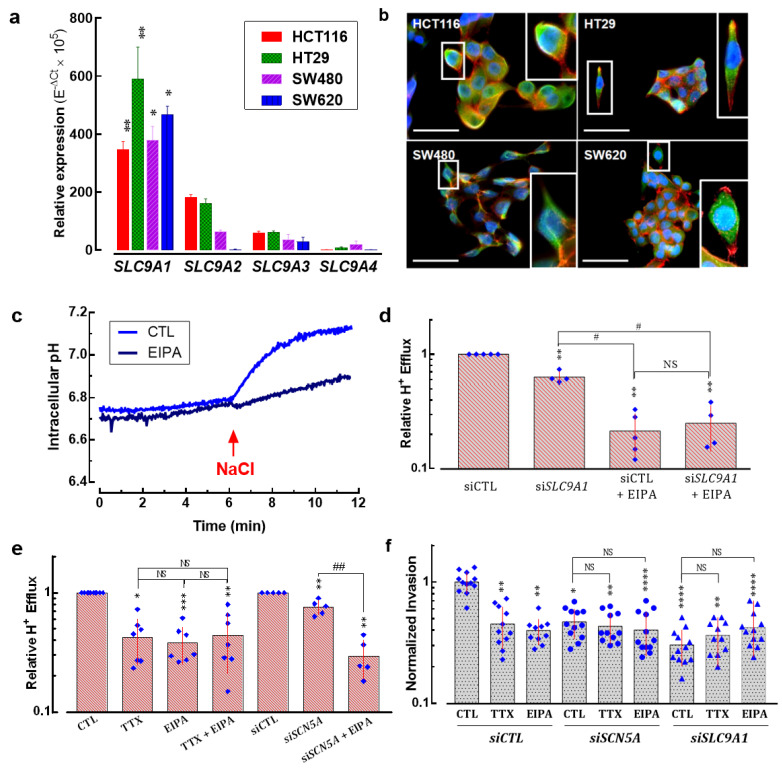
Sodium/hydrogen exchanger type-1 contributes together with Na_V_1.5 channels to colon cancer cell invasiveness. (**a**) Bars represent the relative expression (10^−ΔCt^ × 10^5^) of sodium/hydrogen exchanger family gene members (*SLC9A1*-*SLC9A4*) in HCT116, HT29, SW480 and SW620 colorectal cancer cells. The *SLC9A1* gene (encoding for NHE-1) was the most expressed isoform in these cell lines (*n* = 6). (**b**) Epifluorescence microscopy analysis of the NHE-1 protein in HCT116, HT29, SW480 and SW620 cells. Images show the staining for the NHE-1 protein (green, AlexFluo488), F-actin filaments (red, phalloidin-594) and nuclei (blue, DAPI). NHE-1 protein is mainly located at the plasma membrane of cancer cells. A threefold digital amplification was performed for the cell framed within the white box and shown on the right side of each image. Scale bar 50 µm. (**c**) Representative intracellular pH evolution in NH_4_Cl-acidified SW620 cells resuspended in 130 mM NaCl in the presence of 30 µM TTX or 10 µM EIPA or under control condition. (**d**) Plot shows relative H^+^ efflux measurements after NH_4_Cl-induced intracellular acidification of SW620 cells transfected with *siCTL* or *siSLC9A1* in the absence or presence of 10 µM EIPA (*n* = 5) ** *p* < 0.01 vs. siCTL; ^#^ *p* < 0.05 vs. *siSLC9A1*; NS stands for not statistically different. (**e**) Relative H^+^ efflux of SW620 cells under control condition or transfected with *siCTL* or *siSCN5A* in the absence or presence of 30 µM TTX or 10 µM EIPA (*n* = 5–7). *** *p* < 0.001, ** *p* < 0.01, * *p* < 0.05 vs. control condition (CTL or siCTL); ^##^ *p* < 0.01 vs. *siSCN5A*. (**f**) Effect of silencing the expression of Na_V_1.5 or NHE-1 using specific siRNA (*siSCN5A* or *siSLC9A*, respectively), compared to the transfection of irrelevant siRNA (*siCTL*) on SW620 colon cancer cell invasiveness. These experiments were performed in the absence (CTL) or presence of TTX (30 μM) or EIPA (10 µM). Results are expressed as ratios of mean results obtained with siCTL cells in CTL condition (vehicle). Results are from 10 to 12 independent experiments and were analyzed using Mann–Whitney rank sum tests. **** *p* < 0.0001 compared to CTL condition in siCTL cells; ** *p* < 0.01 compared to CTL condition in siCTL cells; * *p* < 0.05 compared to CTL condition in siCTL cells. NS stands for not statistically different.

**Figure 7 cancers-15-00046-f007:**
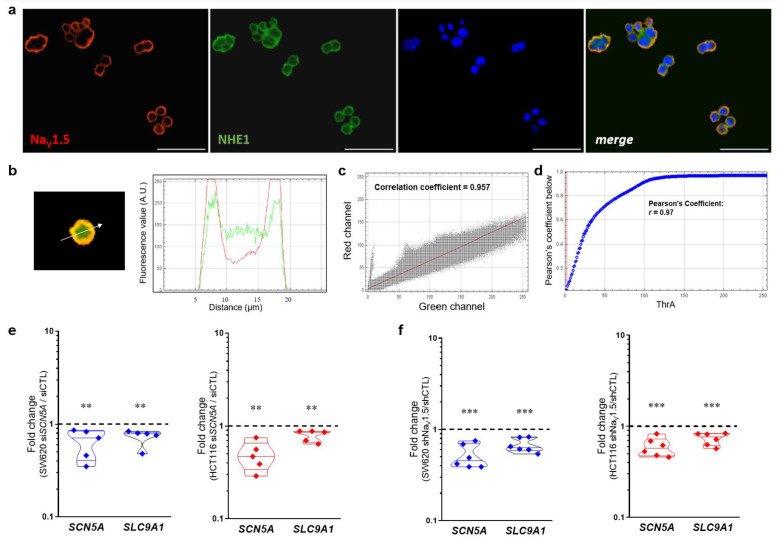
Colocalization analysis of Na_V_1.5 and NHE-1 proteins and regulation of *SLC9A1* transcript expression by Na_V_1.5 sodium channels. (**a**) Epifluorescence microscopy analysis of Na_V_1.5 and NHE-1 proteins in SW620 cells. Images show the staining for Na_V_1.5 proteins (red, detected with the Anti-Sodium Channel Na_V_1.5 antibody produced in rabbit Sigma-Aldrich Ref. S0819), NHE-1 (green, detected with the antibody anti-NHE-1 Santa Cruz Biotechnologies Ref. sc-136239), nuclei (blue, DAPI) and the merge image. Both proteins Na_V_1.5 and NHE-1 are located at the plasma membrane of cancer cells. Scale bar 50 µm. (**b**) A representative example of merging the green and red channels of the epifluorescence images is shown. The right panel shows the fluorescence profile for the two channels along the linear segmentation indicated by the white arrow. (**c**) Cytofluorogram for the fluorescence signals and spatial distribution corresponding to Na_V_1.5 and NHE-1 proteins from images shown in (**a**). (**d**) Plot shows Costes’ automatic threshold analysis of the red and green channels of the images displayed in (**a**), provided a Pearson coefficient of r = 0.97. (**e**) Violin plots show the relative *SCN5A* and *SLC9A1* gene expression in SW620 (left) and HCT116 (right) cells transfected with si*SCN5A* ** *p* < 0.01 vs. siCTL Mann–Whitney rank sum tests. (**f**) Relative *SCN5A* and *SLC9A1* gene expression in shNa_V_1.5 cell lines derived from SW620 (left) and HCT116 (right) cells. Relative analysis was carried out using the shCTL (null-target shRNA) cell lines *** *p* < 0.001 vs. shCTL cell line Mann–Whitney rank sum tests.

**Figure 8 cancers-15-00046-f008:**
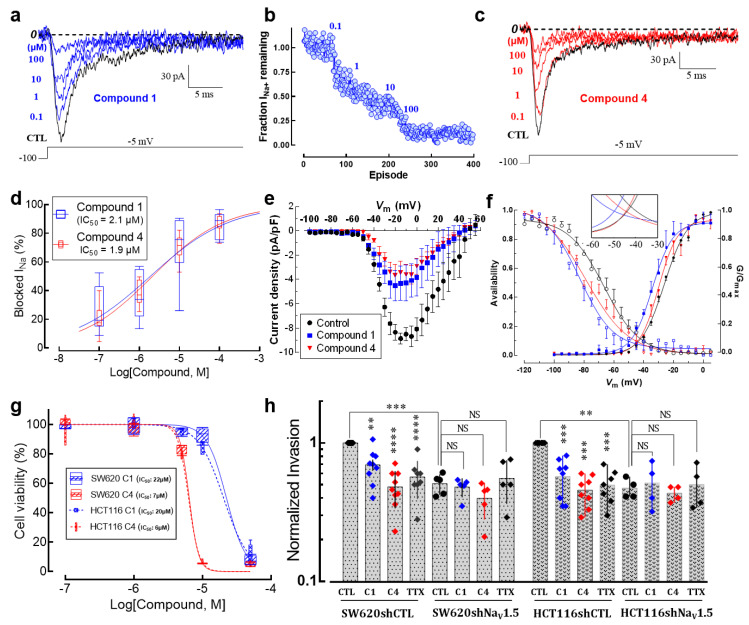
Small-molecule Na_V_1.5 inhibitors reduce 2D colon cancer cell invasiveness. (**a**) Electrophysiological recordings showing the steady-state blockage of Na_V_1.5 currents by increasing concentrations of compound **1** (from 0.1 to 100 µM). Whole-cell patch clamp experiments were carried out in SW620 cells. Voltage steps of 30 ms to −5 mV from an HP of −100 mV evoked Na^+^ currents. The dotted line represents the zero current level. (**b**) Time course of inhibition of Na_V_1.5 currents by compound **1** is shown. Peak currents were normalized to control amplitude (before compound **1** exposure), defined as the fraction of I_Na+_ remaining. Each blue circle means the amplitude at the peak current of an individual trace (episode) normalized to the control current amplitude. The solid line only connects the symbols. Then, the perfusion of the control solution was changed to a solution of the same ion composition plus the lowest concentration of compound **1** (0.1 µM). This solution was continuously perfused until a steady-state block was observed before proceeding to the next concentration and so on. The total duration of this experiment was 400 episodes, meaning around 13 min (1 episode every 2 s). Compounds **1** and **4** were poorly washed out, exhibiting a recovery of the original sodium current <20%. (**c**) Representative recordings of the inhibitory effect of compound **4** on SW620 Na_V_1.5 currents. (**d**) Concentration–response relationships for the effect of compound **1** and **4** on Na_V_1.5 channels. Percentage of Na^+^ current inhibition at each concentration was plotted and the experimental data fitted with a Hill function. Inset shows the IC_50_ values obtained for each compound. (**e**) Current–voltage relationships of Na_V_1.5 channels in the absence and presence of small-molecule compounds. Peak current amplitudes were normalized to the membrane capacitance (*C*_m_) value of each cell. (**f**) Activation (filled symbols) and steady-state inactivation–voltage relationships (empty symbols) of Na_V_1.5 channels in the absence (black circles) and presence of small-molecule inhibitors (C1: blue squares; C4: red triangles). Smooth lines fit a Boltzmann function and *V*_50_ values were calculated for each condition. *V*_50_-activation–voltage values: control −25.9 ± 0.4 mV; compound **1** −34.2 ± 0.9 mV and compound **4** −30.0 ± 0.7 mV, while the *V*_50_−inactivation–voltage values correspond to control −66.6 ± 1.2 mV; compound 1 −71.6 ± 0.9 mV and compound 4 −78.9 ± 1.7 mV. The inset shows the magnified window currents for each condition. Both compounds decreased AUC of the window current; compound **1** reduced about 6% while compound **4** reduced about 30% of the initial window current. (**g**) Effect of small-molecule Na_V_1.5 inhibitors on cell viability of colorectal cancer cells. Plot shows the percentage of viable cells using the MTT technique after 48 h of incubation with different concentrations of compounds **1** and **4** (*n* = 5). (**h**) Summary of SW620 and HCT116 colon cancer cell invasiveness studies performed on Matrigel-coated (0.3 mg/mL) invasion inserts in control condition (0.1% DMSO), in the presence of compound **1** (1 µM), compound **4** (1 µM) or TTX (30 µM). Results from 6–9 independent experiments are expressed relative to the control condition. SW620 and HCT116 cancer cells stably expressing a short hairpin RNA targeting *SCN5A* gene expression (shNa_V_1.5) were also treated under the same conditions. ** *p* < 0.01; *** *p* < 0.001; **** *p* < 0.0001, Mann–Whitney rank sum test.

**Table 1 cancers-15-00046-t001:** Voltage parameters of Na_V_1.5 sodium channels recorded in human colon cancer cells.

Biophysical Parameter	HCT116	HT29	SW480	SW620	Primary Cells
*I*_Na_^+^ _density_ (pA/pF)	−3.3 ± 0.4	−5.4 ± 0.8	−6.3 ± 0.4	−16.8 ± 3.5	−3.1 ± 1.0
*V*_50 activation_ (mV)	−29.6 ± 0.7	−32.1 ± 0.7	−25.7 ± 1.1	−25.6 ± 0.5	−14.3 ± 0.6
*k* _activation_ (ms)	9.4 ± 0.6	8.3 ± 0.6	10.2 ± 0.9	7.6 ± 0.4	8.4 ± 0.5
*V*_rev_ (mV)	58.2 ± 1.3	49.5 ± 1.0	49.8 ± 1.2	54.1 ± 0.9	63.6 ± 5.2
*V*_50 inactivation_ (mV)	−74.6 ± 0.9	−81.5 ± 0.8	−81.7 ± 0.6	−72.9 ± 0.8	−49.5 ± 3.7
*k*_inactivation_ (ms)	11.6 ± 0.8	10.5 ± 0.6	9.2 ± 0.6	10.2 ± 0.7	15.2 ± 3.1
*E*_m_ (mV)	−46.2 ± 4.2	−38.5 ± 3.3	−42.4 ± 3.8	−27.7 ± 6.4	−32.4 ± 8.3
*I*_Na_^+^ _at —10 mV_ (pA/pF)	−3.4 ± 0.35	−5.65 ± 0.68	−6.24 ± 0.95	−16.5 ± 3.34	−2.05 ± 0.63

Data represent mean ± SD. *V*_50_, *k* and *V*_rev_ are given in millivolts and were obtained from *I-V* relationship fits with Boltzmann functions. *V*_50_ and *k* of inactivation were obtained from the fitted data of steady-state inactivation experiments.

**Table 2 cancers-15-00046-t002:** Voltage parameters of SW620-Na_V_1.5 sodium channels in the presence of small-molecule inhibitors.

Biophysical Parameter	CTL	Compound 1	Compound 4
*I*_Na_^+^ _density_ (pA/pF)	−9.7 ± 1.4	−4.1 ± 1.3	−3.6 ± 1.3
*V*_50 activation_ (mV)	−25.9 ± 0.4	−34.2 ± 0.9	−30.9 ± 0.7
*k* _activation_ (ms)	7.8 ± 0.4	6.9 ± 0.8	6.4 ± 0.6
*V*_rev_ (mV)	51.1 ± 2.2	44.4 ± 3.9	43.4 ± 2.3
*V*_50 inactivation_ (mV)	−66.6 ± 1.2	−82.9 ± 1.6	−78.9 ± 1.7
*k*_inactivation_ (ms)	11.8 ± 1.0	11.3 ± 1.3	14.5 ± 1.4

Data represent mean ± SD. *V*_50_, *k* and *V*_rev_ are given in millivolts and were obtained from *I-V* relationship fits with Boltzmann functions. *V*_50_ and *k* of inactivation were obtained from the fitted data of steady-state inactivation experiments.

## Data Availability

All supporting data are available upon request.

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
