# Peer review of "Voltage-Gated Sodium Channel Na_V_1.5 Controls NHE−1−Dependent Invasive Properties in Colon Cancer Cells"

_cancers, 2022, doi:10.3390/cancers15010046_

Round 1

Reviewer 1 Report

General Comments:

 In their manuscript entitled « Voltage-gated sodium Nav1.5 controls NHE-1-dependent invasive properties in colon cancer cells », Lopez-Charcas et al. aimed to highlight the role of both Nav1.5 α-subunit and NHE-1 in the 2D invasion of colon cancer cell lines. The role of both proteins has been already described by the same group in invasiveness properties of breast cancer cells. Here, they further show the functional expression of Nav1.5 channels in cells derived from colon cancer biopsies. Moreover, they provide strong evidence for the effect of new inhibitors of Nav1.5 channel on cancer cell invasion. Despite, the relative lack of novelty regarding the existing literature, the manuscript is well written and the findings are supported by the experimental data. Nevertheless, I have several remarks in order to improve the quality of this work.

Major remarks:

1. The authors show that Nav1.5 and NHE probably regulate cancer colon cell invasion by the same mechanism. They suggest an allosteric interaction as for breast cancer cells. In my opinion, this study could be strengthened by the addition of co-immunoprecipitation and proximity ligation assays experiments to clearly show if there could be an interaction between the two proteins. There are also online tools like LUPIA that could help to predict a possible interaction between proteins based on their aa sequence.

2. The expressions of Nav1.5 and NHE have been compared in tumour tissues to adjacent non-tumoral tissues. Unlike other digestive tissues, it is relatively easy to collect colon biopsies from healthy donors during control colonoscopy procedure. It could be important to study the expression of Nav1.5 and NHE in real non-cancerous colon and possibly to record Nav1.5 currents in cells derived from healthy colon.

3. One of the strengths of this study is the use of primary cell culture derived from biopsy. However, it is not clearly shown if the cells are epithelial cells or fibroblasts. The authors must show that the cells they study are really epithelial cells.

4. The resting membrane potential has been recorded by whole cell patch-clamp but these recordings have not been made under physiological conditions. It is better to use the perforated patch configuration in order to preserve the composition of intracellular media.

5. The cell invasion assay results are shown as normalized to the control. However, it is difficult to really estimate the invasive properties of the cells by this. I could be more informative to also show the number of invasive cells by conditions.

6. The same group already published important results regarding the role of the neonatal isoform of Nav1.5 α-subunit (nNav1.5) in breast cancer cell invasion. In my opinion, the possible presence and role of nNav1.5 must be at least mentioned and discussed in this study.

7. In their conclusion, the authors claim that “This work demonstrates that dysregulation of Nav channels expression is associated with that of NHE-1 exchanger…”. There are no data that clearly support this statement. did the author verify that Nav1.5 silencing regulates NHE expression?

Author Response

Dear reviewer,

We appreciate your critical revision of our work. We have carefully addressed each of your pertinent remarks and present here a revised version of our manuscript that includes experimental evidence of colocalization of NHE-1 and NaV1.5 proteins, as well as in silico analysis of predicted protein-protein interactions and transcriptional-level evidence of possible coregulation of NHE-1 transcripts expression by SCN5A. We have also included evidence from experiments monitoring the epithelial phenotype of the tumor-derived cells used in this study. We present the raw data of the number of invasive cells in the 2D invasion experiments. Finally we have expanded our discussion of the presence of the neonatal NaV1.5 variant, the presence of functional NaV sodium channels in normal intestinal epithelial cells and the reasons why we have used the whole-cell configuration of the patch clamp technique instead of the perforated patch.

Therefore, dear reviewer, we can only thank you for enriching our work.

In the attached document you will find the answers to each of your questions.

Reviewer 2 Report

In this study, the authors found out that NaV1.5 was overexpressed and positively correlated with stage and poor survival prognosis for colorectal cancer patients. Moreover, they found out a differential expression of Nav1.5 and NHE-1 exchanger for tumors originated on the sigmoid colon epithelium. They characterize functional activity of NaV1.5 channels and pharmacology in CRC cell lines and primary cells of colon tumor, and finally they show that the cancer cells invasiveness was reduced by silencing the genes expressing Nav1.5 and NHE-1.

The paper is quite interesting, but in my opinion the authors should thoroughly answer to the following inquiries to have the paper published on Cancers

Major:

The current density of Nav recorded from the various cell lines correlates poorly with the SCN5A expression level: for instance, the latter in SW480 was smaller than the one in HCT116 (of about a factor of 0.6, Fig. 4b), but the current density was about two-fold larger (Table 1): could the authors explain this discrepancy? This discrepancy became very large in the case of colon tumor-derived primary cells (see below).

It would be nice that the authors report in Table 1 the average maximal current (at -10 mV) for every cell line and for colon tumor-derived primary cells, or in Fig. 3 and Fig. 4 legends.

 The late Nav current recordings from colon tumor-derived primary cells show a large outward current, whose size increases with the depolarization amplitude (Fig. 4d) that was not present in the recordings from colorectal cancer cell lines (Fig. 3a). Moreover, the average IV peaked at about 0 mV in the former (Fig. 4e), while it peaked at -10 mV in the latter (Fig. 3f). Finally, the activation and steady-state inactivation-voltage relationships of Nav1.5 channels from colorectal cell lines meet in between -50 and -60 mV (Fig. 3g), while the ones from colon tumor-derived primary cells meet at about -30 mV. All these data indicates that there is another current contaminating the recording from colon tumor-derived primary cells, possibly a Kv one. Indeed, the authors do not use any K-channel blocker, and the intracellular solution contains 130 mM of K. In particular, this outward current might also artefactually decrease the peak amplitude of the Nav current, giving a smaller current density (Table 1). This could explain, although very partially, the similar or even smaller  Nav  current density of colon tumor-derived primary cells in respect to the various cell lines (Table 1), although the former have at least an eight-fold larger SCN5A expression in respect to the latter (Fig. 4b).

 Fig. 7b is not explained and the caption “Time course of inhibition of Nav1.5 currents by compound 1 is shown” does not help at all! Did the figure represent a single experiment, where the cell is repetitively stimulated every 2 s (to -5 mV? for how long?) while applying the drug at first at 0.1 micromoles, then, once the current is stable, the same drug is applied at an higher concentration, and so on? What does it mean the blue line that is barely seen behind the data points? Once compound 1 and compound 2 are removed from the extracellular solution, the current is fully recovered (i.e. the blockade is fully reversible)?

Minor:

Line 204: the immersion of the tissues in 90% ethanol for 2 sec is a quite tough manoeuvre; were the three antibiotics used by the authors not enough to rid of all the bacteria contaminating the biopsy?

Line 299: erratum Smal-Molecule; corrigendum: Small Molecule. Please check carefully the paper for these typographical error; moreover, many phrases are quite wordy, the authors should write them more succinctly.

Author Response

Dear Reviewer,

We appreciate your critical remarks and comments to our manuscript. We have carefully addressed each of your interesting questions about the correlation between messenger and protein expression levels of SCN5A in the different cell lines, also the relevance of showing the mean maximal sodium current at -10mV, the presence of an outward current component in primary tumor-derived cells and the blocking performance of small-molecule inhibitors of NaV1.5. We have also described the rationale for the use of 90% ethanol to remove contaminants from tumor tissue biopsies. Finally, we have corrected typographical errors and reordered some paragraphs.

Thus, dear reviewer, we can only thank you for enriching our work.

In the attached document you will find the answers to each of your questions.

Best regards,

Round 2

Reviewer 1 Report

The authors answered all my questions. thank you.

Author Response

Thank you very much for enriching our work.

Reviewer 2 Report

The authors have satisfactorily answered to all my comments and suggestions and, as far as I’m concerned, the paper is worth to be published in Cancers 

Author Response

Thank you very much for your appreciation of our work.
